# NOD2 deficiency increases retrograde transport of secretory IgA complexes in Crohn's disease

Nicolas Rochereau[1]✉, Xavier Roblin[1], Eva Michaud[1], Rémi Gayet[1], Blandine Chanut[1], Fabienne Jospin[1], Blaise Corthésy[2] & Stéphane Paul [1]

Intestinal microfold cells are the primary pathway for translocation of secretory IgA (SIgA)-pathogen complexes to gut-associated lymphoid tissue. Uptake of SIgA/commensals complexes is important for priming adaptive immunity in the mucosa. This study aims to explore the effect of SIgA retrograde transport of immune complexes in Crohn's disease (CD). Here we report a significant increase of SIgA transport in CD patients with NOD2-mutation compared to CD patients without *NOD2* mutation and/or healthy individuals. NOD2 has an effect in the IgA transport through human and mouse M cells by downregulating Dectin-1 and Siglec-5 expression, two receptors involved in retrograde transport. These findings define a mechanism of NOD2-mediated regulation of mucosal responses to intestinal microbiota, which is involved in CD intestinal inflammation and dysbiosis.

[1] GIMAP/EA3064, Université de Lyon, CIC 1408 Vaccinology, F42023 Saint-Etienne, France. [2] R&D Laboratory of the Division of Immunology and Allergy, CHUV, Centre des Laboratoires d'Epalinges, 1066 Epalinges, Switzerland. ✉email: nicolas.rochereau@univ-st-etienne.fr

The ability of the host immune system to discriminate between pathogens and commensals is essential to maintain mucosal homeostasis[1,2]. The critical importance of maintaining a mucosal homeostatic mechanism in the intestine is highlighted when functional or genetic deficiencies exist. An example of such failure in maintaining a finely balanced immune response is the development of chronic intestinal inflammation, such as Crohn's disease (CD). CD is an idiopathic, chronic regional enteritis that most commonly affects the terminal ileum but has the potential to affect any part of the gastrointestinal tract from mouth to anus. CD is thought to occur as a result of a breakdown in self-recognition of commensal bacteria together with mucosal barrier dysfunction in individuals with a given genetic background[3–5]. The most strongly associated genetic risk factor for CD in Western populations remains NOD2, an intracellular pattern recognition receptor important in immune defense against intracellular microbes[6–8]. NOD2 is known to regulate the intestinal barrier function, limiting the transcellular permeability and bacterial translocation[9,10]. The CD-associated mutation in NOD2 (Leu1007fsinsC, Gly908Arg, and Arg702Trp)[10], located within the LRR region of the protein, results in loss of NF-κB activation in response to muramyl dipeptide (MDP). However, the reasons why the inactivation of NOD2 can result in chronic colitis remain largely speculative.

Secretory IgA (SIgA) is the most abundant immunoglobulin on mucosal surfaces of humans and many other mammals. SIgA can protect the intestinal epithelium by discriminating commensal bacteria from enteric pathogens[11–16]. Recognition of enteric pathogens by the intestinal immune system results in the production of high affinity, T-cell-dependent, pathogen-specific IgA, which is "transcytosed" into the intestinal lumen. SIgA exhibits also the striking feature to adhere to the apical membrane of M cells, promoting the uptake and delivery of antigens (Ags) to dendritic cell (DC) located in Peyer's patches (PP). Under pathological conditions such as infection invading IgA opsonized micro-organisms, these immune complexes amplifies the production of proinflammatory cytokines such as TNF, IL-1β, and IL-23 by human CD103 + DCs[17]. This retrograde transport is called reverse transcytosis, and is mediated by epithelial M cells[18–21]. Both the Cα1 domain of SIgA2 and its associated Sialic acid (Sia) residue glycosylation are involved in IgA reverse transcytosis, as well as Dectin-1 and Siglec-5, identified as receptors for SIgA uptake on M-cells[19]. However, the regulation and pathway(s) whereby SIgA is retro transported across M cells still need to be elucidated.

Increase of the intestinal permeability has for many years been recognized as a pathogenic factor in CD. An abundance of clinical, epidemiologic, and animal model studies have assessed the impact of various commensal and potentially pathogenic enteric bacteria that may trigger or exacerbate IBD[22,23]. In a population-based cohort study, an increased risk of IBD was demonstrated in individuals notified in laboratory registries with an episode of Salmonella gastroenteritis[24]. This finding promotes the concept that pathogens that cause acute intestinal inflammation may predispose individuals to later development of CD, perhaps by causing initial intestinal inflammation or alterations of the intestinal microbiota to promote the formation of colitogenic microbes. We hypothesized that the mucosal inflammation observed in CD patients could be due to an increasing transport of IgA-pathogen complexes from lumen to PP immune cells through M cells. Indeed, after reverse transcytosis, bacteria-IgA complexes are taken up by CD11c+ DCs, and can induce inflammatory responses[18,19]. Moreover, intestinal bacteria selected on the basis of high coating with IgA is associated with reduced gut microbial diversity in human[25] and conferred dramatic susceptibility to colitis in germ-free mice[12,26]. The earliest

observable CD lesions are reported to occur in the follicle-associated epithelium (FAE), where M cells are abundant[27], where the PPs are more numerous, and where IgA2 predominates[28]. NOD2 mutations associated with CD primarily predispose to the development of lesions in the ileal compartment[29], indicating that disease susceptibility is increased by altering signaling interactions between intestinal microbiota and the mucosal innate immunity. Hence, we next hypothesized that NOD2 regulates IgA retrograde transport might explain the dysregulation observed in patients with NOD2 mutations.

Here we observe an increasing transport of IgA in human PP biopsies obtained from CD patients with or NOD2 mutations. We demonstrate that NOD2 has a regulatory effect in the IgA transport through human and mouse M cells by decreasing the Dectin-1 and Siglec-5 expression on M cells, already identified to act as co-receptors in this process. Our findings define a mechanism of NOD2-mediated regulation of innate immune responses to intestinal microbiota which is involved in the initiation and/or perpetuation of the mucosal inflammation observed in CD patients.

## Results

### NOD2 mutation increases SIgA retro transport in CD patients.
We initially examined whether the transport of SIgA was modified as a consequence of the disease, and second refined the analysis to samples collected from CD patients with or lacking NOD2 mutations. To address this question, we first determined NOD2 genotypes from PP biopsies in the terminal ileum from healthy ($n = 8$) or CD ($n = 20$) followed by quantification of SIgA-positive cell numbers. The mean fold-increase of SIgA-positive cells per PP was 2.8 in CD patients expressing NOD2 frameshift polymorphisms ($n = 10$) compared to healthy individuals ($n = 8$) and CD patients expressing wild type (WT) NOD2 ($n = 10$). A first colocalization (Fig. 1b) with anti-IgA Ab and anti-secretory component (SC) Ab confirm that IgA was not enriched from inside (interstitial fluids) but from the lumen (retrotranslocation of SIgA from the lumen to the PP). Colocalization IgA+/SC+ was observed in all tested biopsies (healthy $n = 8$; CD patients with NOD2 frameshift polymorphisms $n = 10$; CD patients with WT NOD2 $n = 10$). A second colocalization IgA2+/DC-SIGN+ (Fig. 1b) indicate that counted SIgA+ cells derived from the retrograde transport of SIgA through M cells followed by their DC uptake. Increase of SIgA retro transport is not observed in segments of the intestine lacking PPs (Fig. 1a, b). Weak differences in numbers of SIgA+ cells were measured in patients with NOD2 polymorphisms R702W ($n = 4$) or FS1007insC ($n = 3$) or R702W/G908R ($n = 3$) (Fig. 1a). The sum of the data demonstrate an increase in PP-associated SIgA in the subset of CD patients displaying mutated NOD2 gene solely.

### NOD2 deficiency increases IgA retro transport in mice.
In the absence of NOD2, PPs present a higher level of CD4+ T cells and M cells in the FAE and increased levels of Th1 (IFN-γ, TNF-α and IL-12) and Th2 (IL-4) cytokines. These immune alterations are associated with an increase of paracellular permeability and yeast/bacterial translocation[30,31]. Hence, we next hypothesized that at steady-state, the dysregulation observed in CD patients with NOD2 mutations may favor SIgA reverse transcytosis, and ultimately lead to excessive uptake of bacteria-SIgA complexes[32].

To this aim, we sought to determine whether the loss of NOD2 expression affected the transport of SIgA in a NOD2 knock-out (KO) mouse model. Transport of fluorescently labeled SIgA was compared between NOD2 KO mice and WT mice by examining the fate of the antibody molecule after administration into a ligated intestinal loop containing a PP. As observed in human gut

biopsies obtained from patients with NOD2 polymorphisms, the mean fold-increase (2.3) of SIgA transport from the lumen into the PP in *NOD2* KO mice (112.4 ± 14.5) was significantly different when compared to WT mice (49.4 ± 7.6). This significant increase in IgA transport, together with a significant decrease after MDP-PAM treatment demonstrates the implication of *NOD2* in the process of transcytosis (Fig. 2a). Finally, we

did not observe any retrotranscytosis of negative controls as BSA or an irrelevant mouse IgG using a ligated intestinal loop model suggesting that the effect is SIgA-dependant (Fig. 2a).

Oral administration of HIV p24 in association with SIgA to littermate WT mice leads to the generation of a robust immune response against the viral antigen. In this experimental setting, SIgA serves as a delivery system carrying an intact HIV p24

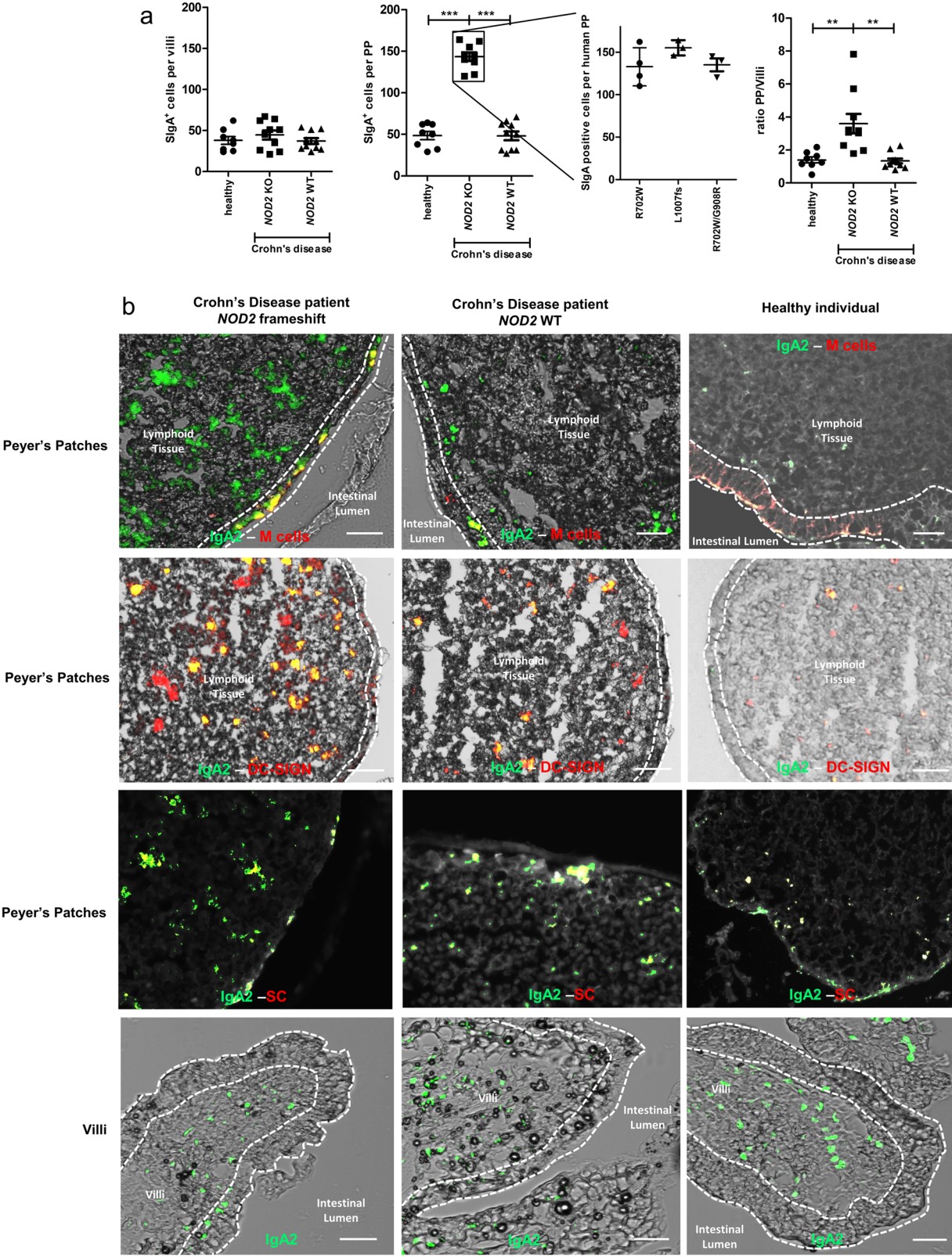

**Fig. 1 NOD2 mutation increases SIgA reverse transcytosis in CD patients. a** The two first graph compares the frequency of IgA-positive cells between healthy donors (n = 8), CD WT NOD2 patients (n = 10) and CD patients expressing NOD2 polymorphisms (n = 10) in PP and in villi. The third graph shows the number of IgA positive PP cells in patients expressing different NOD2 polymorphisms. NOD2 polymorphisms (R702W, G908R, FS1007insC) were determined by qPCR and sequencing. The last graph shows the ratio PP/villi. Vertical bars show the mean value ± SEM. A nonparametric Mann–Whitney U-test was used (p values: **p < 0.01, ***p < 0.005). **b** Images obtained from patient biopsy samples taken from the terminal ileum. This experiment was repeated in all patients with similar results: healthy donors (n = 8), CD WT NOD2 patients (n = 10) and CD patients expressing NOD2 polymorphisms (n = 10). Biopsies were labeled with anti-human GFP-IgA2 and PE-GP2 to label M cells or PE-DC-SIGN or PE anti-human SC at room temperature for 2 h. The left panels show representative images from CD patients expressing NOD2 polymorphism, the middle panels, CD patients with WT NOD2, and the right panels depict images from healthy donors. Bars: 200 μm. On all pictures, dotted lines delineate the follicle–associated epithelium (FAE) separating the intestinal lumen and the lymphoid tissue (side view). Source data are provided as a Source Data file.

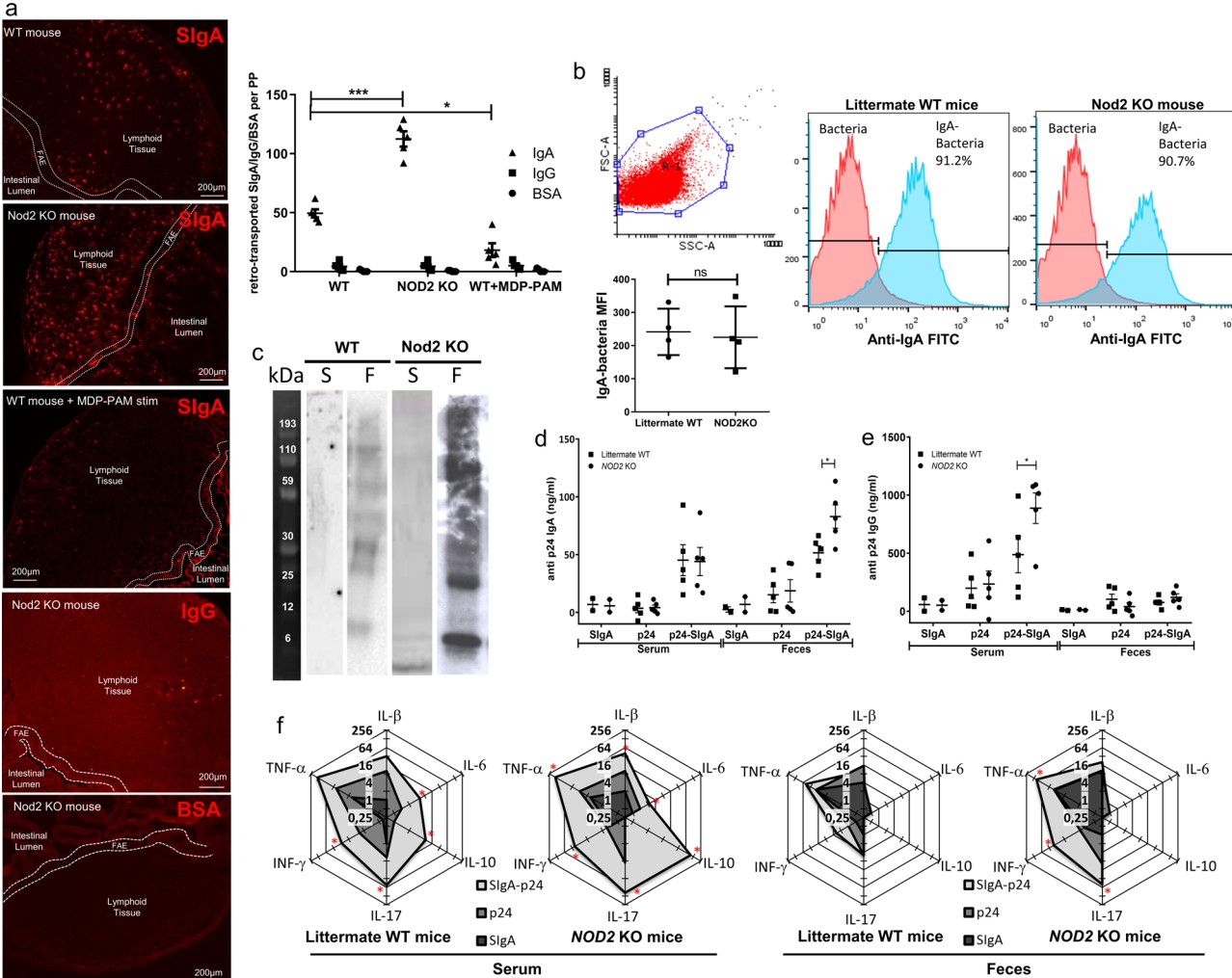

**Fig. 2 NOD2 deficiency increases IgA retrograde transport in mice. a** Tissue section showing a PP obtained from a ligated intestinal loop in WT, NOD2 KO or WT mice after MDP-PAM stimulation following exposure to SIgA-Cy3 for 60 min. Mouse IgG and the BSA were used as negative controls. The graph shows the number of SIgA-positive cells per PP. This experiment was repeated on 6 mice per group and an average of 3 PP per mouse was done. Vertical bars show the mean value±SEM. One-way ANOVA followed by Bonferroni post hoc test was used (p values: *p < 0.05; ***p < 0.005). **b** Fecal bacteria from WT or Nod2KO mice were stained with anti-IgA-FITC and the MFI were calculated on IgA-bacteria. This experiment was repeated on 4 mice per group. Vertical bars show the mean value ± SEM. A nonparametric Mann–Whitney U-test was used (ns: not significant). **c** Lysed bacteria were used as the target antigens in a western blot, using serum (S) or fecal (F) supernatant from WT or NOD2 KO mice as primary antibody and anti–IgA-HRP as the secondary antibody. This experiment was repeated on eight cohoused mice (WT (n = 4), NOD2 KO (n = 4)) with similar results. NOD2 KO and littermate WT mice were cohoused and immunized orally with p24-SIgA. **d** Levels of Ag-specific mucosal IgA and **e** levels of serum IgG in littermate WT or NOD2 KO mice were shown. Vertical bars show the mean value ± SEM. n = 5 biologically independent mice per group. One-way ANOVA followed by Bonferroni post hoc test was used (p values: *p < 0.05; **p < 0.01). **f** Cytokine concentrations in serum and faeces were determined in triplicate by Luminex-multiplex cytokine assay in littermate WT (n = 3) and NOD 2 KO mice (n = 3). Cytokine profiles are shown as radar charts; each axis displays the mean quantity (pg/ml) of each cytokine after immunization with p24-SIgA, p24 and PBS. A nonparametric Mann–Whitney U-test was used. P values have been calculated by comparing the p24-SIgA group with the p24 group; *p < 0.05. Source data are provided as a Source Data file.

antigen into intestinal M cells in PPs[20]. We thus speculated that the increased retro-transport of SIgA observed in *NOD2* KO mice would result in improved p24-specific reactivity after oral immunization as compared to littermate WT mice. First, we confirmed the same ability of IgA to bind to the same microbiota pattern in littermate or *NOD2* KO mice, as previously described[33] (Fig. 2c). It was also confirmed by quantification of IgA$^+$ bacteria with flow cytometry (Fig. 2b). As oral immunization is well known to induce both systemic and mucosal responses, the antigen-specific response was measured in both serum and feces samples 1 week after the last immunization. Systemic p24-specific IgG and fecal IgA titers were significantly increased in *NOD2* KO mice as compared to littermate WT mice (Fig. 2d, e). Oral delivery of an equivalent amount of p24 alone was not sufficient to induce a specific Ab responses in either KO or littermate WT animals (Fig. 2d, e). Because the local tissue-dependent nature of immune responses relies highly on the controlled production of cytokines, a panel of them was measured by Luminex in serum and feces obtained 24 h after the last immunization. In comparison with p24 alone, p24-SIgA was a potent inducer of all the tested cytokines in serum but also TNF-α, INF-γ and IL-17 measured in feces of *NOD2* KO mice (Fig. 2f). Immunization with SIgA alone has no effect on the production of cytokines. Except for IL-10, no difference in cytokine secretion was observed between littermate WT and *NOD2* KO mice. The increase of IL-10 in serum after immunization with p24-SIgA in the *NOD2* KO mice may indicate regulation of the systemic inflammatory process following initial priming of the specific Th cell response. Taken together, these results indicate that transport of p24-SIgA complexes is dependent on functional *NOD2* and results in the regulated passage of the hooked Ag, which is subsequently processed to trigger the onset of mucosal and systemic antibody and cytokine responses.

**SIgA-*Salmonella* amplify *Salmonella*-induced colitis in mice.** The demonstration that SIgA reverse transcytosis in *NOD2* KO mice favors the transport of a bound Ag, as reflected by the detection of specific immune responses, suggests that such a mechanism may be involved in the pathogenesis of chronic colitis. To address this hypothesis, KO mice displaying opposed effects on intestinal SIgA retrograde transport (i.e., *NOD2* KO mice identified as promoting intestinal retrograde transport (this study), and *Dectin-1* KO mice known to abrogate reverse transcytosis[19]) were used and compared with WT littermate animals. The disease condition was triggered upon administration of dextran sodium sulfate (DSS) in drinking water (positive control), or after oral delivery of *Salmonella* Typhimurium alone (*Salmonella*-induced colitis model) or in association with murine *Salmonella*-specific SIgASal4 (working hypothesis). In control experiments, oral administration of the Dectin-1 antagonist laminarin, known to suppress the development of DSS-colitis[34], was used to further verify the impact of SIgA retrograde transport inhibition.

In comparison with the PBS control group, oral administration of *Salmonella* alone in littermate WT mice resulted in a significant increase of inflammation severity, as determined by the measurement of the disease activity index (DAI) (Fig. 3a). Oral delivery of *Salmonella*-SIgASal4 complexes led to a weak, yet statistically significant further increase in DAI at day 5 (Fig. 3a). DSS-treated littermate WT mice showed the highest signs of disease. Blocking of Dectin-1, and thus reduction of SIgA reverse transcytosis by addition of laminarin in the drinking water, markedly decreased the severity of colitis. In support of our working hypothesis, *NOD2* KO mice with increased SIgA reverse transcytosis and thus higher intestinal transport of *Salmonella*

exhibited a DAI resembling that of DSS-treated animals (Fig. 3a), well above the index observed in the *Salmonella* alone condition. Such a colitis-preventing effect was impaired upon delivery of laminarin to mice (Fig. 3a). The "clinical" progression of colitis was also confirmed by quantifying neutrophil infiltration in the *lamina propria* (Fig. 3b), according to the Nancy histological score (Fig. 3c), by measuring weight loss (Supplementary Fig. 1a), serum IL-6, CRP, and Lipopolysaccharide (LPS) (Supplementary Fig. 1b). The high level of blood LPS in *Salmonella*-IgA treated mice reflects the IgA-based retrotranscytosis of *Salmonella* through the intestinal epithelium. This seems to demonstrate the inability of Sal4 IgA to neutralize *Salmonella* under our experimental conditions as previously described[35]. To confirm this point, we measured the aggregation of *Salmonella* Typhimurium after SIgASal4 binding[35]. We did not observe in vitro a significant increase in *Salmonella* aggregates after SIgA binding (Supplementary Fig. 1c). The absence of aggregates confirms the non-neutralizing character of Sal4 IgA in our experimental conditions. This could indicate that these aggregates are only capable of forming in vivo and do not alter the retrotranscytosis ability of IgA.

SIgA-*Salmonella* treated mice without laminarin have been isolated, plotted and significantly highlighted by comparing littermate WT and Nod2KO mice (Supplementary Fig. 1d). The significative difference of DAI at day 5 after SIgA-*Salmonella* administration clearly confirms the role of Nod2 in SIgA-*Salmonella* transport. This statement was also observed using a *Salmonella*-GFP bound with SIgA administered into a ligated intestinal loop containing a PP in *NOD2* KO mice and WT littermate mice (Supplementary Fig 1e). Results obtained by blocking Dectin-1 by the addition of laminarin in the drinking water were similar by using *Dectin-1* KO mice (Supplementary Fig. 1f) and are consistent with the previous reports[34]. Thus, SIgA reverse transcytosis seems to play a role in colitis induction by a *NOD2*-mediated increased transport of pathogens.

**NOD2 modulates the Dectin-1 and Siglec-5 expression.** In the mouse intestine, selective IgA reverse transcytosis across the epithelium has been shown to depend on the expression of Dectin-1 and to a lesser extent Siglec-5 on M cells[19]. IgA reverse transcytosis can be recapitulated in the in vitro model of FAE comprising human polarized Caco-2 cells and M-like cells. Traditional transport (transcytosis) across polarized epithelial of serosal polymeric IgA into secretions requires polymeric Ig receptor and a plethora of evolutionary well-conserved intracellular proteins including EEA-1, Rab-5, Rab-9, Rab-11, Rab-17, and Rab-25. Because the regulation of vesicle trafficking along the retro-transcytotic pathway is unknown, we found it important to determine whether *NOD2* may somehow contribute to fine-tune the expression of either one of the two receptors and/or the putative organelle-specific proteins.

In order to address the nature of proteins possibly involved, we took advantage of the in vitro model of FAE. In the absence of M-like cell conversion, the plain Caco-2 cell polarized monolayer (mono-culture) allowed weak retrotranscytosis of the reporter IgA-luciferase fusion protein (IgA-Luc), in agreement with previous data[19]. This background level was not affected after transfection with siRNA targeting a selection of extra- and intracellular proteins, as indicated in Fig. 4a. In the presence of interspersed M-like cells in the polarized monolayer (co-culture), SIgA reverse transcytosis of IgA-Luc was increased by a factor of 4 (Fig. 4a, lanes control and random non targeting oligo). Among the candidate assayed, targeting of EEA-1, Rab-5, Rab-17, as well as SYK and TAK1 proteins involved in Dectin-1 signaling pathway led to a significant 3-fold decrease of SIgA-Luc

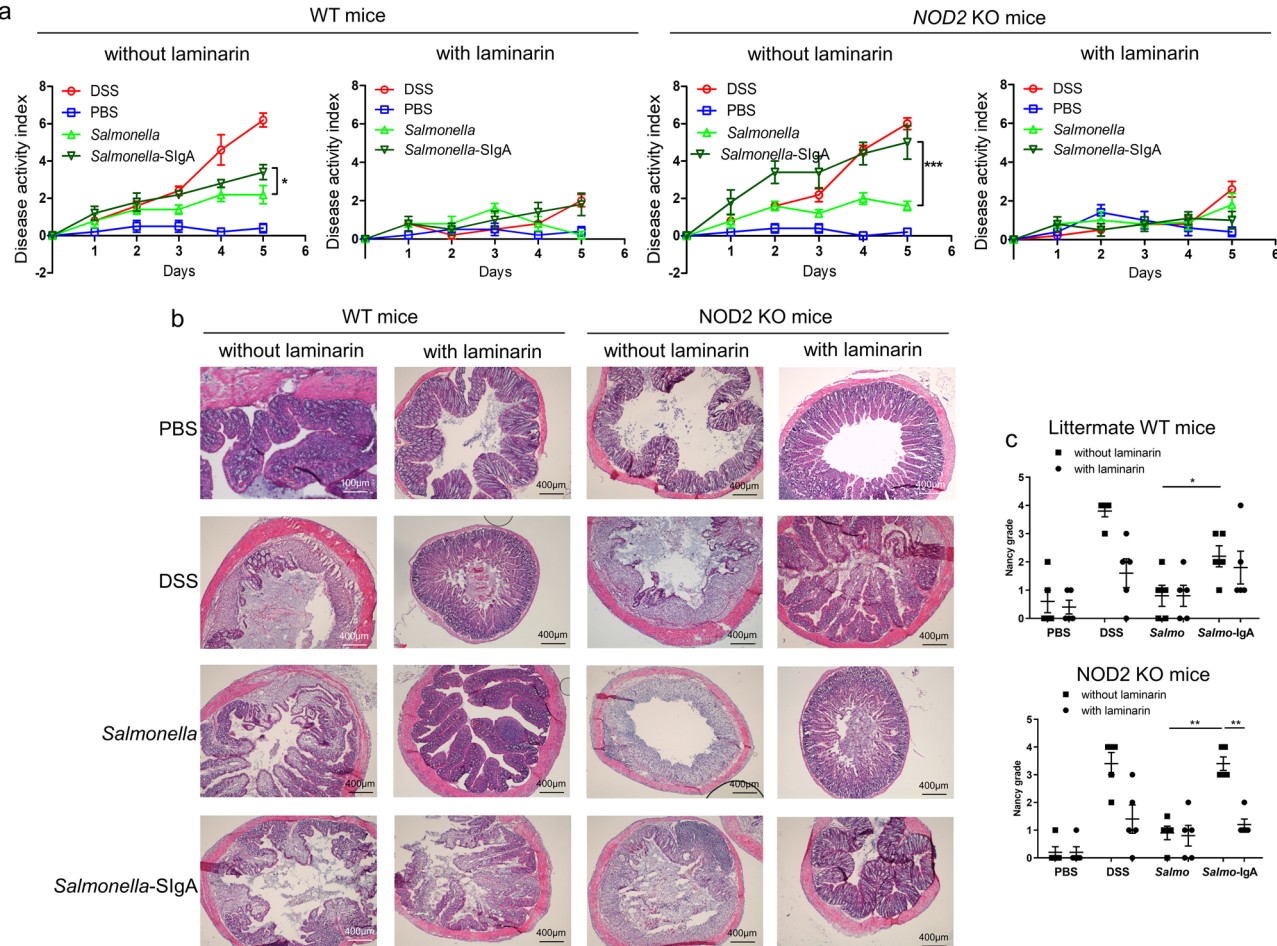

**Fig. 3 SIgA-Salmonella amplify Salmonella-induced colitis in mice.** Cohoused mice were challenged orally either with DSS, PBS, *Salmonella* Typhimurium or *Salmonella* Typhimurium bound with murine IgA. They were left untreated or treated with 5% laminarin in drinking water for 3 days prior to colitis challenge. **a** Disease activity index (DAI) score was undertaken daily to evaluate the clinical progression of colitis. The DAI was the combined score of weight loss compared to initial weight, stool consistency, and bleeding. Vertical bars show the mean value±SEM. $n = 5$ biologically independent mice per group. One-way ANOVA followed by Bonferroni post hoc test was used ($p$ values: $*p < 0.05$; $***p < 0.005$). **b** Sections from the colon of mice showing neutrophil infiltrates in the lamina propria. This experiment was repeated on all mice showing similar results. **c** Nancy histological score applied in each colon section for littermate and *NOD2* KO mice. $n = 5$ biologically independent mice per group. One-way ANOVA followed by Bonferroni post hoc test was used ($p$ values: $**p < 0.01$). Source data are provided as a Source Data file.

transport, as compared to controls (Fig. 4a). This decrease compares with that resulting from the targeting of Dectin-1 and Siglec-5 used as positive controls. A significant twofold increase of IgA transport in cells treated with *NOD2* siRNA, together with a significant threefold decrease after MDP-PAM treatment confirm the implication of *NOD2* in the process of transcytosis, and indirectly validates the results obtained via the FAE model (Fig. 4a). Western Blot (Supplementary Fig. 2a) or quantitative RT-PCR was performed to correlate the decrease of mRNA level and its expected increase after MDP/PAM treatment, (Supplementary Fig. 2b). In additional control experiments, both the decrease in the level of targeted proteins and the monolayer integrity (TEER) were systematically monitored after siRNA transfection (Supplementary Fig. 2c). No significant modifications between controls, knockdown and stimulated cells were observed. Staining of tight junctions using ZO-1 detection confirmed the monolayer integrity after siRNA transfection (Supplementary Fig. 2d). Finally, the retro transport of cholera toxin (CT), which is known to use the Rab-5 pathway[36], has been using as a positive apical-to-basal transport control to test the efficacy of siRNA knockdown on either monoculture or coculture conditions (Supplementary Fig. 2e). After Rab-5 siRNA

knockdown, CT transport is significantly reduced in our inverted in vitro model, confirming the role of Rab-5 in CT transport and the efficacy of siRNA knockdown.

To identify if EEA-1, Rab-5, Rab-17, Dectin-1, and Siglec-5 are key partners in IgA reverse transcytosis, IgA2 was added apically in the in vitro model of FAE for 30 or 60 min, and IgA2-associated proteins complexes were recovered after immunoprecipitation with protein M-agarose beads. Association between IgA2 and EEA-1, Rab-5, Rab-17, Dectin-1, and Siglec-5 was revealed by immunodetection with a battery of specific antibodies (Fig. 4b). Specific colocalizations IgA[+]/EEA-1[+], IgA[+]/Rab-5[+], and IgA[+]/Rab-17[+] were also confirmed by immunofluorescence. No colocalization was observed with Rab-7, Rab-9, Rab-11, Rab-25, and pIgR with IgA (Fig. 4c).

The role of *NOD2* in the modulation of protein expression associated with SIgA reverse transcytosis was investigated next. The in vitro model containing M-like cells (co-culture) or not (mono-culture) was used as such, or following of *NOD2* knockdown with siRNA. Proteins were recovered from whole cell lysates and their relative abundance was assessed by Western blot (Fig. 5a). The expression of Dectin-1 and Siglec-5 was significantly increased after *NOD2* knockdown, whereas no

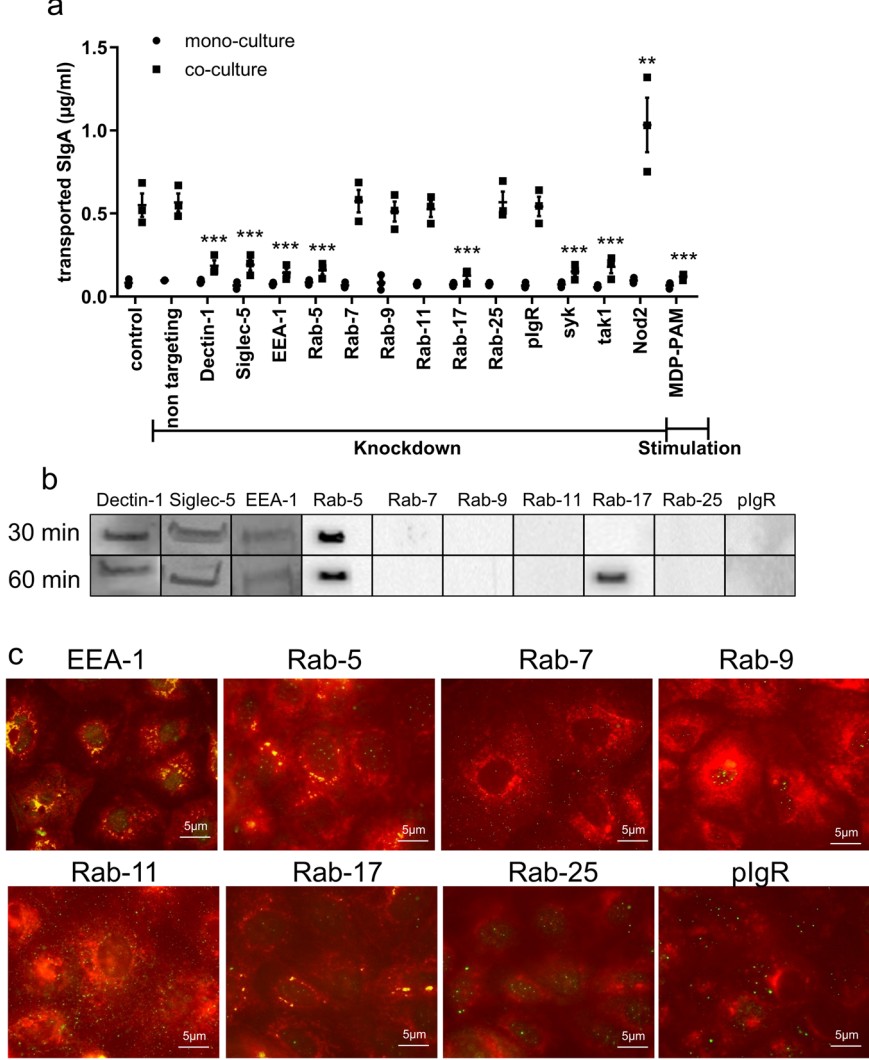

**Fig. 4 IgA reverse transcytosis after protein inhibition or stimulation.** After siRNA knockdown or muramyl dipeptide—Pam3Cys (MDP-PAM) stimulation, IgA2 conjugated with luciferase **a** transport were quantified in the inverted in vitro model of FAE. Vertical bars show the mean value ± SEM. $n$ = 3 independent experiments. One-way ANOVA followed by Bonferroni post hoc test was used ($p$ values: **$p < 0.01$; ***$p < 0.005$). **b** Immunoprecipitation made 30 or 60 min after IgA incubation. This experiment was repeated twice with similar results. **c** Immunofluorescence staining after IgA incubation (60 min), with FITC anti-human IgA and anti-Rabs mAbs or anti-EEA-1 mAbs followed by corresponding PE secondary Abs. Colocalization between IgA and endosomal proteins resulted in yellow dots present in EEA-1, Rab-5, and Rab-17 images. This experiment was repeated twice with similar results. Source data are provided as a Source Data file.

relevant changes were observed upon analysis of the endosomal proteins tested. MDP-PAM treatment led to a decrease of the expression of the two IgA receptors (Fig. 5b). When examining specifically the expression of Dectin-1 and Siglec-5 in GP2+ M-like cells, as tested by flow cytometry, a consistent threefold increase of either receptor that occurred in *NOD2* knockdown cells was confirmed as compared to unconverted cells with an enterocyte phenotype (Fig. 5c). In support of data in Fig. 5b, MDP-PAM treatment did not result in any significant changes in the expression of either receptor. Dectin-1 and/or Siglec-5 expression has not been detected on enterocytes. This observation has been confirmed in NOD2 KO mice where Dectin-1 expression is increased on PP M cells compared to WT mice (Fig. 5d). Finally, we investigate the involvement of NOD2 in other stages of the mucosal IgA pathway, such as production. Figure 5e clearly shows no significate difference in IgA concentration observed in serum or feces between NOD2KO mice and littermate WT mice. The sum of the data implicates NO*D2* in M-like cells as the plausible player involved in Dectin-1

and Siglec-5-mediated facilitated retrograde transport of IgA across a tight epithelial mimic.

## Discussion

So far, the mechanisms involved in the loss-of-function polymorphisms on downstream *NOD2* signaling and the pathogenesis of CD remain largely unknown. A commonly recognized finding is that defects in *NOD2* result in a constitutively weak inflammatory response that can lead to increased intestinal bacterial load and with time to chronic intestinal inflammation observed in CD[8,37]. Here we demonstrate that the mucosal inflammation observed in CD patients is could be due to increased transport through M cells of IgA-bacteria complexes from the lumen to immune cells present in the PP. First, we found that IgA reverse transcytosis was significantly increased through human (Fig. 1) and mouse (Fig. 2) M cells when *NOD2* is mutated or absent. At a steady-state, *NOD2* seems to down-regulate the IgA retrograde transcytosis through M cells. In CD or colitis mouse models, the

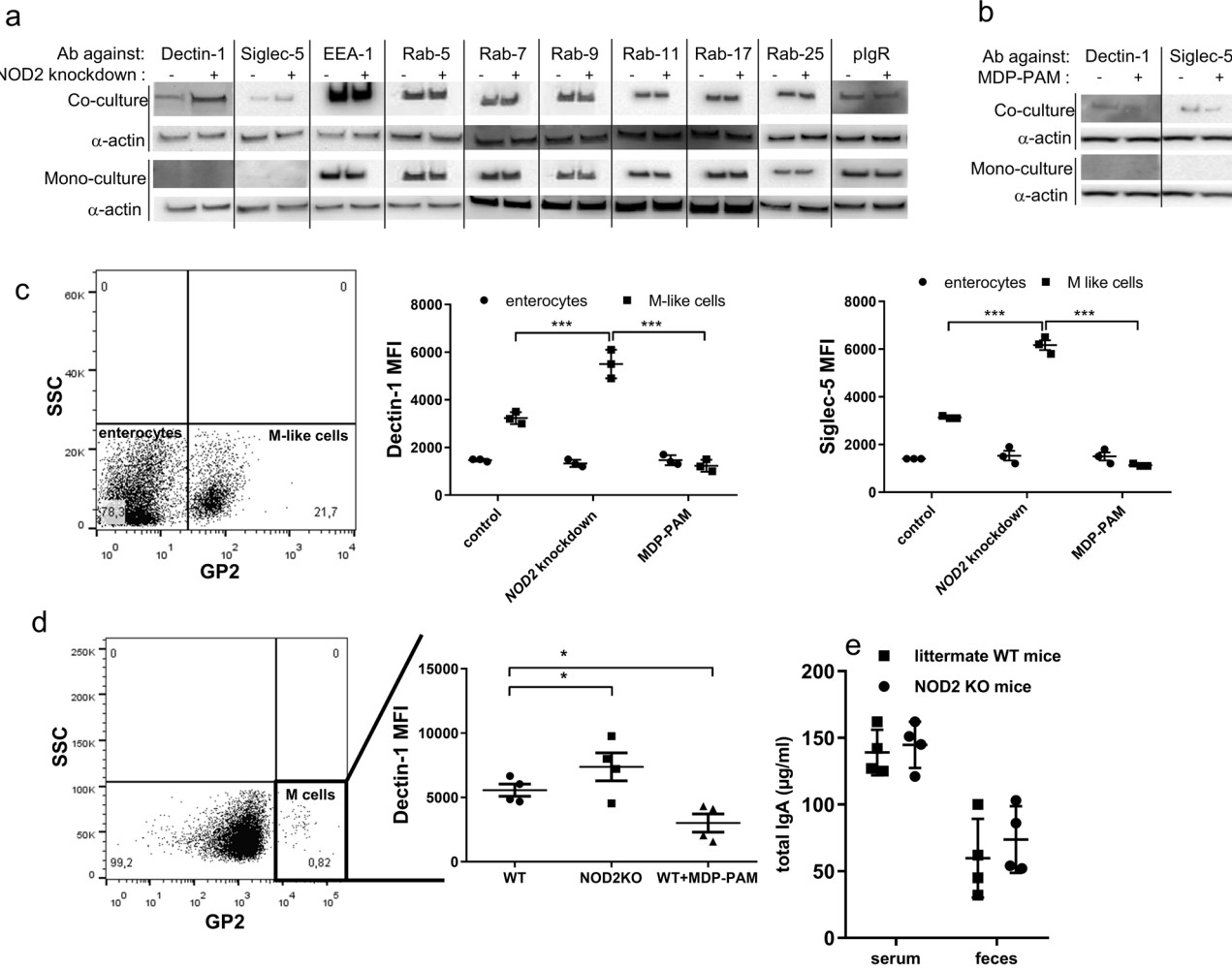

**Fig. 5 NOD2 modulates the expression of Dectin-1 and Siglec-5 receptors.** Western blot showing the expression of proteins in an in vitro model containing (co-culture) or not (mono-culture) M-like cells before "−" or after "+" NOD2 blocking of transcription with siRNA knockdown (**a**) or NOD2 stimulation with MDP-PAM treatment (**b**). These experiments were repeated twice with similar results. Flow cytometry was used to examine the role of NOD2 on Dectin-1 and Siglec-5 expression in M like cells in vitro (**c**) ($n = 3$ independent experiments; Vertical bars show the mean value ± SEM; One-way ANOVA followed by Bonferroni post hoc test was used, p values: ***$p < 0.005$) and in vivo (**d**) ($n = 4$ independent experiments; Vertical bars show the mean value ± SEM; A nonparametric Mann–Whitney U-test was used, p values: *$p < 0.05$). **e** Total IgA concentration was determined by ELISA in serum and faeces of NOD2KO mice and littermate WT mice. $n = 4$ biologically independent mice. One-way ANOVA followed by Bonferroni post hoc test was used. Vertical bars show the mean value ± SEM. kd knockdown, stim stimulation. Source data are provided as a Source Data file.

absence of *NOD2* could increase the transport of IgA-bacteria complexes inducing mucosal inflammation. It is well established that the polymorphism of the *NOD2* gene is a major risk factor in CD. However, a molecular explanation of how such loss of function leads to increased susceptibility to CD remains unclear. Hedl et al.[38] have shown that NOD2 signaling activates the mTOR pathway which induces the upregulation of anti-inflammatory mediators and simultaneously the downregulation of pro-inflammatory cytokines. These data support the idea that *NOD2* modulates innate immune responses to intestinal microflora and thus suggest that the absence of such regulation leads to increased susceptibility to CD.

The role of the IgA reverse transcytosis in the pathogenesis of chronic colitis was identified in Dectin-1 KO mice and littermate WT mice using the well-known *Salmonella* colitis model[39,40]. Dectin-1 signaling has been described to regulate intestinal inflammation by controlling commensal Lactobacillus-mediated colonic regulatory T cells[34]. It seems that the transport of IgA-bacteria complexes through M cells via Dectin-1 receptor[19] induces intestinal inflammation, suggesting some degrees of

correlation between the abundance of M cells and chronic intestinal inflammation. In this respect, Bennet et al.[41] found that in both the dextran sodium sulfate and *Citrobacter rodentium* models of colitis, significantly increased numbers of PP M cells were induced at the peak of inflammation in the colonic epithelium. Using KO mice, we demonstrate that *NOD2* acts as an immune regulator of IgA reverse transcytosis. Indeed, *NOD2* KO mice administered with *Salmonella*-IgA complexes exhibit an increase of the inflammation's severity as compared to littermate WT mice. This observation is supported by studies showing that NOD2 can modulate inflammation and mediate efficient clearance of bacteria from the mucosal tissue during *Salmonella* colitis[42,43].

We next investigated the molecular mechanisms which could explain how *NOD2* modulates IgA retrograde transport. Using a specific siRNA knockdown approach in the in vitro model containing M-like cells, the involvement of *NOD2* as a regulator of IgA transport was confirmed. The role of endosomal proteins already known to be involved in the IgA transport through enterocytes was also studied[44]. Our data reveal that IgA reverse

## IgA Retrograde Transport

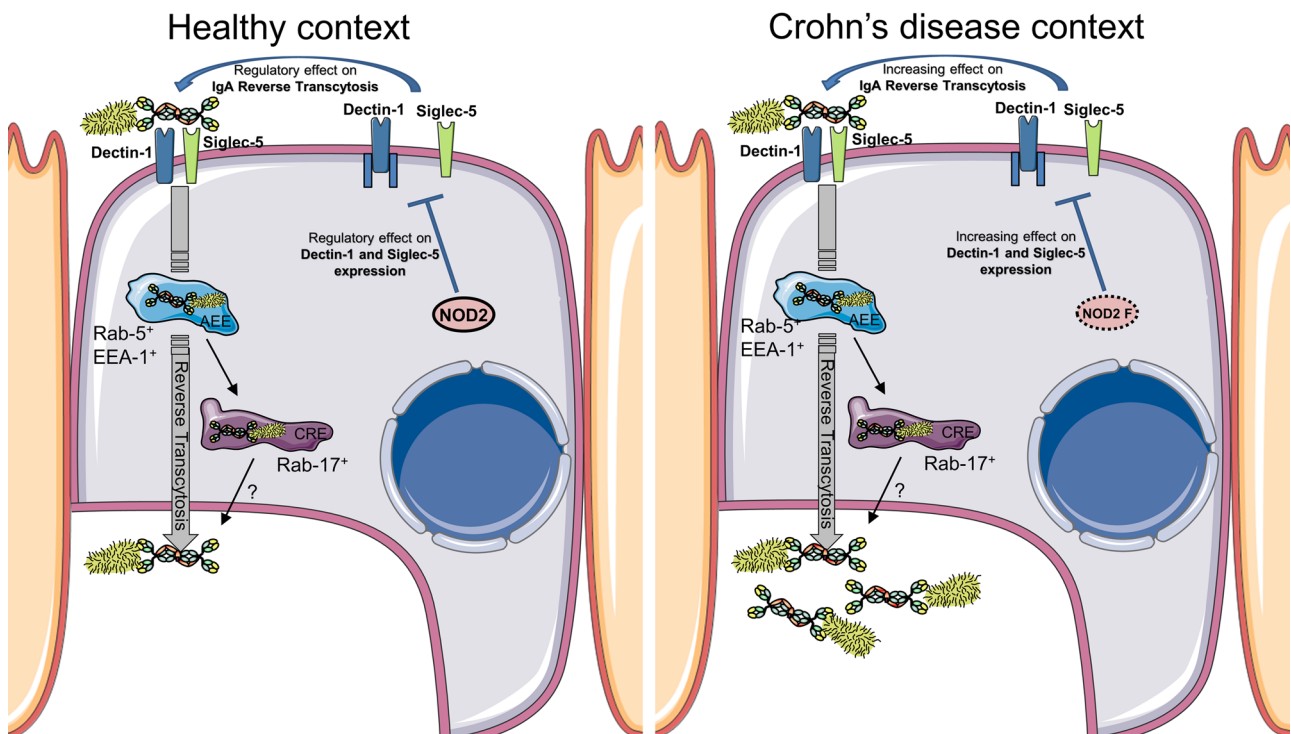

**Fig. 6 The retrograde transport of IgA-pathogen complexes across M cells is increased in CD patients as compared to healthy individuals.** This translocation of IgA is regulated by WT NOD2 upon decrease of Dectin-1 and Siglec-5 expression in M cells, ensuring proper homeostasis. This pathway is controlled by endosomal proteins such as EEA-1, Rab-5, and Rab-17. We would like to emphasize that this identified mechanism is likely one among others that is involved in the initiation and/or perpetuation of mucosal inflammation observed in CD patients.

transcytosis through M cells seems to be mediated by EEA-1, Rab-5, and Rab-17 endosomal proteins as it has been previously described for epithelial IgA transcytosis. The role of Dectin-1 and Siglec-5 in IgA reverse transcytosis was also confirmed[19]. NOD2 has also been shown to influence MHC cross-presentation[29], autophagy induction, and resistance to intracellular bacterial infection[45,46]. Thus, while principally well-known for its acute signaling effects, NOD2 activation causes a variety of cellular changes in vivo that are also likely important for immune homeostasis. SYK and TAK1 proteins from the Dectin-1 signaling pathway are also involved in the blocking of IgA reverse transcytosis. The role of NOD2 interaction with TAK1 through its leucine-rich repeat (LRR) region to exert its inhibitory effect on TAK1-induced NF-κB activation has been published[47]. This suggests that NOD2 inhibits TAK1-induced NF-κB activation, which results in the downregulation of IgA transport. A significant decrease of Dectin-1 and Siglec-5 expression after NOD2 knockdown in M cells is observed in the current study, suggesting that NOD2 contributes to regulate the expression of these two receptors for SIgA (Fig. 6). Of interest in the more global context of intestinal diseases, NOD2 has been identified as a negative regulator of TLR2[48] and of TLR4 in necrotising enterocolitis[49].

Our finding should be interpreted with caution, as the etiology of CD involves a combination of genetic, environmental, and microbial factors[37]. Hence, we find it fair to underline that the SIgA-dependent route identified in the study is likely to be one among other mechanisms that eventually initiates and/or perpetuates mucosal inflammation observed in CD. In any case, our data confirm and extend the knowledge that the NOD2 genotype status is currently the strongest genetic marker associated with a severe CD course. In Fig. 1, NOD2 mutations of the studied population

(R702W, 1007 fs and R702W/G908R) were compared and no differences were observed in IgA-positive cells distribution in PP. These results are consistent with the study of Hugot et al., which provided strong evidence that the penetrance of the most at-risk genotypes is low. They found no clear relationship between mutation frequencies and the disease incidence in their studied populations and no significant deficit of double-dose mutation carriers among healthy controls[50]. However, another study predicts an 8% increase in the risk for complicated disease with the presence of a single NOD2 mutation, and a 41% increase with 2 mutations[51]. Nevertheless, these two studies confirm that CARD15/NOD2 acts in interaction with other unknown risk cofactors.

A theory proposes that antibodies generated in response to microbial colonization of the intestine shape the microbiota composition to benefit the health of the host through a process called antibody-mediated immunoselection (AMIS)[52]. Immunoselection refers to a process of natural selection within a host organism that is mediated by the immune system to influence microbial fitness and hence microbial ecology and evolution. A significant fraction of commensal bacteria is heavily coated with IgAs[26,53,54]. However, such coating seems to be non-specific, as there is a significant overlap of bacterial species between IgA-coated and non-coated fractions. Under such conditions, the bacteria diversity was significantly reduced. Thus, it seems that reduced affinity maturation of IgAs is associated with reduced diversity and skewed microbiota and abundant coating of bacteria with natural IgAs. A recent paper goes further by saying that bacteria recognized by human SIgM were dually coated by SIgA and showed increased richness and diversity compared to IgA-only-coated or uncoated bacteria[55]. This bacterial selection mechanism could be a causal agent for CD development or other

IgA-based diseases where IgA reverse transcytosis could play a role. Indeed, CD patients have several features in common with IgA nephropathy and celiac disease. Increased small intestinal mucosal permeability has been demonstrated in these pathologies[56,57]. In IgA nephropathy, this may allow the influx of food and bacterial Ags resulting in immune complex formation and deposition. In celiac disease, the alteration of para- and transcellular pathways has been proposed to explain the retrograde transport of intact peptides, and notably the apico-to-basal translocation of SIgA ensured by the transferrin receptor[58].

A better understanding of molecular mechanisms driving chronic gut inflammation has led over the past two decades to therapeutic strategies with major impacts for the current management of IBD[59,60]. However, despite spectacular successes, mainly attributable to the anti-TNF therapy, not all patients respond to the drugs and about one third of the responders relapse within a short period of time. Further works are therefore needed to identify therapeutic molecules in IBD. In the present study, we demonstrated that IgA retro-transport is involved in promoting inflammation in CD by acting on the transport of IgA-bacteria immune complexes in the PP through M cells. Therapeutic strategies aiming at blocking IgA reverse transcytosis during the acute phase of CD may be considered to design new efficient immunotherapeutic strategies.

## Methods

**Immunolabeling of human PP ex vivo.** Informed and consenting CD patients or healthy individuals who had undergone lower endoscopy for routine diagnostic purposes with normal intestinal mucosa provided two biopsy samples from the terminal ileum. Biopsies were fixed for 2 h in 3% paraformaldehyde and included in OCT embedding solution, before being cryosectioned using a Leica cryostat model CM1950. Seven-micrometers sections were captured on Ultra+ Superfrost microscope slides and stained for M cells. Slides were washed in PBS to eliminate residual OCT embedding solution, and blocked with PBS containing 5% FBS for 30 min at room temperature. Immunolabeling was performed using a combination of GFP-IgA2 (Invivogen), anti-human PE-GP2 mAb (MBL), anti-human PE-DC-SIGN mAb (ThermoFisher scientific), and mouse anti-IgA secretory component Ab (Abcam) followed by goat anti-mouse PE-IgG (Abcam) diluted to 20 µg/ml for 2 h at room temperature. The slides were then washed in PBS, air-dried, and mounted with Fluoprep (Biomérieux). Slides were observed by immuno-fluorescence microscopy (Eclipse, Nikon).

**Oligonucleotide sequences for NOD2 polymorphism PCR.** PCR of NOD2 polymorphisms[61] (R702W, G908R, FS1007insC) were performed on DNA extracted from human biopsies with a commercial extraction kit (QIAamp DNA mini kit, Qiagen, Hilden, Deutschland). The following primers were used to identify specific mutations: R702W: forward, 5′-GAA TTC CTT CAC ATC ACT TTC CAG T -3′ and reverse, 5′-GTC AAC TTG AGG TGC CCA ACA TT-3′; G908R: forward, 5′-CCC AGC TCC TCC CTC TTC-3′ and reverse, 5′-AAG TCT GTA ATG TAA AGC CAC-3′; FS1007insC: forward, 5′-CTG AGC CTT TGT TGA TGA GC-3′ and reverse, 5′-TCT TCA ACC ACA TCC CCA TT-3′. We purified PCR products with a PCR purification kit (Qiagen) before sequencing (Eurofins).

**Mice.** NOD2 KO mice were obtained from Gabriel Nunez (University of Michigan, USA). Dectin-1 KO mice were obtained from Gordon D. Brown (University of Aberdeen, UK). NOD2 KO mice[62] and Dectin-1 KO mice[63] have been described. Littermate mice were obtained from Nod2-heterozygous crosses. All mice were born and hosted at the PLEXAN (Platform for Experiments and Analysis, Faculty of Medicine, Université de Saint-Etienne, France) which is a conventional animal facility with infectious sector P2. All mice were co-housed in the same conditions (Temperatures of 20 °C with 50% humidity, 12 light/12 dark cycle, unlimited access to food and water), and were females between 2- and 4-months old. The experimental protocols have been approved by the French ministry of research, the local ethical committee (CEEA-Loire) and the Animal Welfare Committee of the PLEXAN (agreement no. 2017011315316714_v4).

**SIgA administration into ligated loops.** For ileal loop preparation, mice were starved overnight, anesthetized by intra-peritoneal injection of a mix of ketamine and xylazine (100 and 10 mg/kg animal weight, respectively) and kept warm at 37 °C throughout the surgical procedure. Hundred microliters of a 1 mg/ml solution of SIgA-Cy3 diluted in PBS or Salmonella(GFP)-SIgASal4 immune complexes (see a model of mouse colitis), BSA (Sigma), or mouse IgG (anti-human TAK1 mAb (R&D System) were administered into a 1.5-cm ileal loop containing a PP. Salmonella enterica subsp. enterica serovar Typhimurium GFP were obtained from ATCC (14028GFP™). Upon completion of the experiment, the mice were sacrificed

by cervical dislocation and the piece of intestine was removed, extensively washed with PBS, fixed for 2 h in 3% paraformaldehyde, and included in optimal cutting tissue (OCT) embedding solution. Seven-micrometers sections (Leica cryostat model CM1950, Leica Microsystems) were captured on Ultra+ superfrost microscope slides (VWR International). Slides were observed by immunofluorescence microscopy (Eclipse, Nikon). The protocol followed the guidance of the regional Ethics Committee for animal testing, CREEA (Permit number: No. 69387487).

Mice stimulated by MDP-PAM were administered with MDP-PAM in the ligated loop, two hours before SIgA-Cy3 administration.

To measure mouse SIgA retrotranscytosis, a polymeric IgA Ab (clone IgAC5 specific to S. flexneri serotype 5a LPS[64]) was obtained as previously described[65]. Purified free human SC was produced in Chinese Hamster Ovary cells[66]. SIgA was obtained by mixing in PBS pIgA molecules with a twofold excess of human SC for 2 h at room temperature according to the conditions described previously[67]. Cy3-SIgA complexes were obtained by conjugation with indocarbocyanine (Cy3) using the FluoroLink mAb Cy3 labeling kit (Amersham Biosciences).

*Study of IgA/fecal microbiota interactions.* The ability of fecal IgA to bind gut microbiota has been measured by western blot[33]. Lysed bacteria were used as the target antigens in a western blot assay. Serum or fecal supernatant from WT or Nod2KO mice were used as primary antibody and anti-IgA-HRP as secondary antibody. This experiment was repeated on 4 mice per group. The quantity of IgA already bound with fecal bacteria were also quantified by flow cytometry. Fecal supernatant from WT and Nod2KO mice were stained with anti-IgA-FITC and the MFI were calculated on IgA-bacteria. This experiment was analysed using 4 mice per group.

**In vivo delivery of p24-SIgA.** HIV-1 p24 capsid protein from clade B strain (Px Therapeutics, France) was covalently associated with polymeric SIgAC5 using the Sulfo-KMUS heterobifunctional crosslinker (Thermo Scientific). Covalent complex formation was verified by Western blot with a polyclonal anti-HIV-1 serum and revealed with anti-human IgG HRP-conjugated secondary Ab (Amersham). Mouse oral immunizations were performed by orogastric intubation with polyethylene tubing under light anesthesia with isofluroan (Halocarbon Laboratories). The tubing was introduced at a fixed distance of 1.8 cm from the incisors. Immunizations consisted of three administrations of 100 ml at 1-week intervals. Littermate WT ($n = 5$), and Nod2 KO ($n = 5$) mice were immunized with 100 mg of HIVp24-SIgA, or SIgA alone, or HIVp24 alone per administration.

**Measurement of HIVp24-specific IgG and IgA Abs.** Serum and feces samples were recovered 1 week after the last immunization. Five fresh feces were collected from each animal. Feces were incubated with Halt Protease Inhibitor Cocktail (Thermo Scientific), centrifuged at $16,000 \times g$, and stored at $-20$ °C until use. Specific Abs against HIVp24 were measured using a quantitative ELISA. Maxisorp 96-well plates were coated with either 50 µl of HIVp24 Ag solution (5 µg/ml in sterile PBS) or 50 µl of a 1/3200 dilution of an equal mixture of anti-mouse Ig kappa and lambda light chain–specific mAbs (Serotec), and then incubated O/N at 4 °C. Murine IgG or IgA immunoglobulins (Igs) (Southern Biotech) were used as standards. Bound or captured Igs were detected by incubation with HRP-conjugated goat antimouse (IgG), while IgA was detected using biotinylated goat antimouse IgA (Southern Biotech) followed by streptavidin-HRP (Amersham). Results are given as the means of concentrations ±SEM.

**Cytokines and chemokines.** The evaluation of multiple cytokines/chemokines was performed with a Luminex 100 instrument (Luminex Corporation, Austin, TX, USA), in combination with the Bio-Plex mouse cytokine 23-plex panel and Bio-Plex mouse cytokine Th17 panel B 8-Plex Group III (Bio-Rad, Berkeley, CA, USA). Biological fluids were recovered 1 week after the last immunization. Cytokine and chemokine concentrations were determined as the mean of three replicates.

**Model of mouse colitis.** The virulent streptomycin-resistant Salmonella enterica serovar Typhimurium strain SL1344 was cultured in LB (LB Broth, Sigma) supplemented with 90 µg/ml of streptomycin (LB-St). A day before infection, a SL1344 colony was cultured overnight at 37 °C, 100 rpm in 3 ml of LB-St. Mouse IgASal4, specific for Salmonella Typhimurium surface carbohydrates was produced as described previously and used to treat mice with SIgA-Salmonella complexes. In other experiments, colitis was also induced by giving 5% dextran sodium sulfate solution in drinking water ad libitum for 7 days[68]. Body weight and DAI were monitored each day.

NOD2 KO ($n = 5$), Dectin-1 KO ($n = 5$), and littermate WT ($n = 5$) mice were given either Salmonella-SIgASal4 complexes or Salmonella alone (i.e., $10^6$ CFU/mouse) in 100 µl PBS by orogastric intubation with polyethylene tubing under light anesthesia with isoflurane (Halocarbon Laboratories). Mice were not pre-treated with streptomycin as we would like to measure the role of SIgA reverse transcytosis in the context of a normal microbiota. Infectivity and dissemination of Salmonella SL1344 were fist tested at different doses. A dose of $1 \times 10^6$ CFU/mouse was used as it was not lethal but induce strong inflammation.

Induction of colitis was compared in NOD2 KO ($n = 5$), Dectin-1 KO ($n = 5$), and littermate WT ($n = 5$) mice after adding 5% laminarin in drinking water for 3 days before colitis induction[34].

**Assessment of colitis severity**. During the duration of the experiment, the DAI score was monitored daily to evaluate the clinical progression of colitis[69]. The DAI score combines read-outs including: weight loss compared to initial weight, stool consistency, and rectal bleeding. Scores were defined as follows: weight loss: 0 (no loss), 1 (1–5%), 2 (5–10%), 3 (10–20%), and 4 (>20%); stool consistency: 0 (normal), 2 (loose stool), and 4 (diarrhoea); and rectal bleeding: 0 (no blood), 2 (visual pellet bleeding), and 4 (gross bleeding, blood around anus). The experimental endpoint was reached when mice exhibited weight loss >20% of initial weight with dehydration and diarrhoea had to be euthanized by cervical dislocation following inhalation of isoflurane.

To evaluate histological damages reflecting colitis severity, a small fragment (0.5 cm) of the colon was cut, embedded in OCT, and frozen in isopentane cooled with liquid nitrogen. Seven-micrometer sections were prepared as previously described and stained with hematoxylin/eosin using the published procedures[70]. Scoring of neutrophil infiltration was performed using the Nancy histological score which has been evaluated by a blinded pathologist.

Mice were bled 5 days after treatment through retro-orbital plexus. The presence of IL-6, LPS, and CRP in sera was assessed by ELISA (Mouse IL-6 ELISA MAX, Biolegend, San Diego, CA; Mouse Lipopolysaccharide ELISA Kit, ELISAgenie, London, UK; Mouse C-Reactive Protein/CRP DuoSet ELISA, R&D system, USA).

**IgA-*Salmonella* agglutination assay**. *Salmonella* aggregates formed by IgASal4 has been quantified before oral challenge[35]. To test the in vitro agglutination of S. typhimurium by IgASal4 antibodies, 0.1 ml of hybridoma culture supernatants (1 µg) was added to 0.1 ml of an overnight culture of bacteria and incubated in round-bottom ELISA plates. Unrelated IgA hybridoma supernatant (anti-V. cholera[71]) or fresh culture medium was used as a control. Agglutination was measured by flow cytometry after 3 h at 23 °C.

**Cell culture**. The human intestinal cell line Caco-2 (clone 1) (obtained from Dr. Maria Rescigno, University of Milan-Bicocca, Milan, Italy)[72] was cultured in Dulbecco's modified Eagle's medium (DMEM) (PAA) supplemented with 10% (v/v) fetal bovine serum (FBS, Thermo-Fisher), 1% (v/v) non-essential amino-acids (PAA), and 1% (v/v) penicillin-streptomycin (PAA), referred to as complete DMEM. The human Burkitt's lymphoma cell line Raji B (American Type Culture Collection), was cultured in RPMI 1640 supplemented with 10% (v/v) FBS, 1% (v/v) non-essential amino-acids, 1% (v/v) L-glutamine and 1% (v/v) penicillin-streptomycin.

**Inverted in vitro model of the human FAE**[19,73]. Inverted Transwell polycarbonate inserts (12 wells, pore diameter of 3.0 µm, Corning) were coated with Matrigel™, a basement membrane matrix (BD Biosciences) prepared in pure DMEM to a final protein concentration of 100 µg/mL for 1 h at room temperature. The coating solution was removed and inverted inserts were washed with 300 µl of DMEM. Caco-2 cells ($3 \times 10^5$), suspended in 300 µL of complete DMEM, were seeded on the lower insert side and cultured overnight. The inserts were then inverted and placed in a 12-well culture dish and kept for 9 days. Raji B cells ($5 \times 10^5$) were resuspended in complete DMEM and then added to the basolateral compartment of the Caco-2 cells, and co-cultures were maintained for 5 days. Mono-cultures of Caco-2 cells, cultivated as above but without back-addition of Raji B cells, were used as controls. The establishment of polarized co- and mono-cultures was controlled by measurement of TEER using an EndohmTM tissue resistance chamber (Endohm-12, World Precision Instruments) connected to a Millicell-ERS Ohmmeter (Millipore). The mean TEER value of medium alone ($9 \, \Omega/cm^2$) was subtracted from each measurement. In random samples, the barrier integrity of the tight junctions was confirmed by zonula occludens-1 (ZO-1) immunolabeling[74]. For transcytosis analyses, inserts were inverted prior to incubation in a 6-well plate, and a piece of silicon tubing (14 (height) × 20 (diameter) mm, Labomoderne) serving as a medium reservoir was placed on the surface facing the basolateral pole of the cell monolayer.

**Gene inhibition by small interfering (si) RNA**. Cells in the inverted in vitro model of FAE were transfected at a final concentration of 5 nM with ON-TARGETplus SMARTpool siRNAs (Dharmacon) using Silentfect reagent (Bio-Rad) according to the procedure provided by the manufacturer. The reference numbers for gene targeting were as follows: Dectin-1: L-021476-00-0005; Siglec-5: L-019522-02-0005; EEA-1: L-004012-00-0005; pIgR: L-017729-00-0005; Rab-5: L-004009-00-0005; Rab-7: L-010388-00-0005; Rab-9: L-004177-00-0005; Rab-11: L-004726-00-0005; Rab-25: L-010366-00-0005; Syk: L-003176-00-0005; *NOD2*: L-003464-00-0005. TAK-1 was silenced by using sequence 6317S (Cell Signaling Technology).

**Treatment with MDP - Pam3Cys (MDP-PAM)**. In order to specifically activate *NOD2*, the inverted in vitro model of FAE was exposed to 1 µg/ml MDP and 1 µg/ml Pam3Cys-Ser-(Lys)4 hydrochloride (Invivogen) for 24 h.

**Immunoprecipitation and Western Blot**. 10 µg of IgA2 (Invivogen) were added to the apical compartment of the in vitro model of FAE. And the cells were incubated for 30 or 60 min at 37 °C. After two washes with PBS, the cells were lysed with Mammalian Protein Extraction Reagent (Thermo Scientific). The lysate was cleared by centrifugation and the protein concentration was brought to 5 µg/ml. IgA2 (and

bound proteins) were concentrated by immunoprecipitation with protein M-agarose beads (Invivogen). Post washes, elution was performed with 0.1 m glycine (pH 3.0), with immediate neutralization by 1 M Tris buffer (pH 8.0). The eluted material was subjected to SDS-PAGE followed by transfer onto a hybond ECL nitrocellulose membrane (GE Healthcare Life Science). Immunodetection of targeted proteins was performed with a selection of Abs/antisera including: Goat anti-human Dectin-1/CLEC 7A serum, anti-human CD170 (Siglec-5) mAb (mouse IgG1, Clone #194128), sheep anti-human EEA-1 serum, anti-human pIgR mAb (Mouse IgG3, Clone # 825724), anti-human SYK mAb (mouse IgG1, Clone # 720402), anti-human TAK1 mAb (mouse IgG1, Clone # 491840) were purchased from R&D System, and rabbit anti-human Rab-5 serum, anti-human Rab-7 mAb (mouse IgG2b, Clone # Rab-7-117), anti-human Rab-9 mAb (mouse IgG1, Clone # Mab9), rabbit anti-human Rab-11 serum, anti-human Rab-25 mAb (Rabbit IgG, Clone EPR18353) were obtained from Abcam. Appropriate HRP-coupled secondary Abs were used for detection with the "Clarity Western ECL Substrate" (Biorad).

*Immunofluorescence staining*. Ten micrograms of IgA2 (Invivogen) were added to the apical compartment of the in vitro model of FAE. The cells were incubated for 60 min at 37 °C. Inserts were washed in HBSS to eliminate residual medium, incubated in 4% paraformaldehyde for 30 min, permeabilized with 0.1% Triton X-100 (Sigma-Aldrich), and blocked with PBS containing 5% FBS for 15 min at room temperature. Immunolabeling was performed using a combination of same anti-rabs and anti-EEA-1 mAbs described in the previous section. Each reagent was diluted to 1/100 and incubated for 2 h at room temperature. Corresponding secondary antibodies labeled with a PE were incubated for 1 h at room temperature. After two washes, inserts were air-dried, mounted with Fluoprep (BioMerieux), and observed by Immunofluorescence microscopy (Eclipse Ti, Nikon).

***NOD2* RT-qPCR**[75]. Total RNA was extracted using TRIZOL (Invitrogen). Reverse transcription was performed using the PrimeScript RT reagent kit (TaKaRa Biotechnology, Dalian, PRC). The SYBR Premix Ex Taq™ II kit (TaKaRa Biotechnology) was used to amplify *GAPDH* and *NOD2* gene products. The oligonucleotide primers used were as follows: *NOD2*: forward, 5′-CTG AAG AAT GCC CGC AAG GT-3′ and reverse, 5′-GTC TCT TGG AGC AGG CGG ATG-3′; *GAPDH*: forward, 5′-TGC ACC ACC AAC TGC TTA GC-3′ and reverse, 5′- GGC ATG GAC TGT GGT CAT GAG-3′. The double standard curve method was used to analyze the relative gene expression[76].

**IgA RT experiment**. After 48 h of siRNA mediated gene knockdown or 24 h of MDP-PAM stimulation, 10 µg of Ab conjugated with luciferase (Luc) or colostrum IgA or 1 µg of CT from V. cholerae (sigma) were added to the apical side of the in vitro model of FAE at 37 °C for 90 min[19]. Basolateral solutions were then recovered and the number of retro-transcytosed Ab-Luc measured by luminometry using the Gaussia Luc Assay Kit (Biolux) according to the procedure provided by the manufacturer. Transported colostrum IgA and CT were detected by ELISA using respectively biotinylated goat anti-human IgA (Southern Biotech) and biotinylated rabbit anti-CT pAb (Invitrogen) followed by streptavidin-HRP (Amersham).

**Statistical analysis**. Statistical analyses were performed using the InStat version 2.01 from the GraphPad Software. A nonparametric Mann–Whitney $U$-test or one-way ANOVA followed by Bonferroni post hoc test was used where appropriate[77,78]. The limit of significance for $p$ values was set at 0.05 (marked by * in the plot); ** indicates $p$ values ≤ 0.01, and *** stands for $p$ values ≤ 0.005. Statistically significant differences between groups are emphasized by bars connecting the relevant columns under comparison.

**Reporting summary**. Further information on research design is available in the Nature Research Reporting Summary linked to this article.

## Data availability
The authors declare that all data supporting the findings of this study are available within the paper and its supplementary information files. Source data are provided with this paper.

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

## Acknowledgements

N. Rochereau was supported by a post-doc fellowship from the Sir Jules Thorn Charitable Trust and subsequently by a post-doc fellowship from Region Rhone-Alpes and also from ANRS (France). The laboratory of S.P. is supported by research grants from ANRS, Sidaction and MSD Avenir, and that of B.C. by grant No. 3100-156806 from the Swiss Science Research Foundation. We would like to thank the staff of the "Unité Hospitalo-Universitaire d'expérimentation animale" technical platforms of IFR143 for help in sample analysis.

## Author contributions

N.R., E.M., R.G., B.C., and F.J. carried out the experiment. N.R. wrote the manuscript with support from X.R., B.C., and S.P. B.C. and S.P. supervised the project.

## Competing interests

The authors declared no competing interests.
