## [Peer Review File · Nature Communications]

Reviewers' comments:

Reviewer #1 (Remarks to the Author):

Rochereau et al. NOD2-dependent influence of secretory IgA complexes in inducing GALT inflammation in Crohn's disease. In the manuscript by Rochereau et al., the authors investigated whether NOD2 plays a role in IgA retrograde transport and whether it may influence the mucosal inflammation involved in Crohn's disease. As NOD2 is the strongest genetic risk factor for CD, this data is of interest to researchers studying host-microbe interactions and their involvement in the etiology of IBD. The authors identified a higher number of SIgA+ cells within ileal PP biopsies of CD patients with NOD2 mutations, as compared to CD without NOD2 mutations and healthy controls. This is the most intriguing result and initiated the remainder of the study investigating whether NOD2 is involved in IgA-complex reverse transcytosis. The IgA complexes co-localized with M cells and DCs, indicating the path via M cells and then taken up by dendritic cells within the PPs. The authors then demonstrated that a similar result could be observed in Nod2^{-/-} mice. The authors use an HIV protein model to further indicate a role for Nod2 in this IgA-complex immune response and while there was increased antigen-specific fecal IgA and serum IgG in Nod2^{-/-} mice, no cytokine differences (except for IL-10) were observed between WT and Nod2^{-/-} mice suggesting no difference in immune response.

The authors nicely show an effect of the response to Salmonella-IgA complexes as compared to DSS and Salmonella challenge, but it is not clear what the strength of the Nod2 involvement is since the Nod2KO mice were not statistically compared to the WT littermate mice receiving the same treatment. Were the WT and Nod2KO mice co-housed or separated at weaning? If the effect of the Salmonella-SIgA is similar in both WT and Nod2KO mice (as it appears in Fig 3), then it would be difficult to determine the exact influence of NOD2 on this phenotype.

The in vitro data showing increased SIgA transport in NOD2 knockdown cells and cells stimulated with MDP, provide strength to the initial findings of increased IgA+ cells in CD patients. Further, increased Dectin-1 and Siglec-5 expression in the knockdown cells and M-cells reinforce the involvement of NOD2 in IgA retrograde transport.

Overall, the findings in CD patients with NOD2 mutations and data in Figure 5 present convincing evidence of a role for Nod2 in IgA-complex transport. This data would add to the field and should be considered for publication.

However, some questions remain:

NOD2 is an intracellular receptor – so would deletion in epithelial cells only (using a villin-cre mouse) lead to the same phenotype? Or does NOD2 need to be deleted in all cells involved in the IgA retrograde transport – e.g. also in plasma cells, dendritic cells?

Since the mice used were littermates, it would be important to confirm that the gut microbiota is not significantly different in the mice used in Fig 2 to show increased IgA transport into the PP. Alternatively, comparing the reactivity of the IgA between WT and Nod2KO mice to the gut microbiota (similar to what was done in McCarthy et al. 2011 JCI) may further indicate when the role of NOD2 is most critical.

Major points:

1. The initial findings and data in Figure 5 present convincing evidence of a role for NOD2 in IgA-complex retrograde transport, however, the in vivo data is underwhelming and does not provide direct evidence of a role for Nod2 in the sensing or immune response to these IgA-complexes. As this is the title and main conclusion of the paper, it is a concern. Could the data from Fig 2 be represented to show WT and Nod2KO mice on the same plot to allow for statistical comparison? This is the only way to clearly show a role for Nod2 in the mice.

Minor point:

1. What is the reference for line 106-107? Is this only in mice, or also in humans deficient for NOD2?

Reviewer #2 (Remarks to the Author):

In this manuscript, Rochereau and colleagues investigate the role of NOD2 in influencing the retrotranslocation of SIgA and SIgA-immune complexes into mouse (and a limited number of human) Peyer's patch tissues. While the study is intriguing, the investigators are quick to draw conclusions without sufficient experimental numbers and/or without important controls to justify claims about specificity of the SIgA transport in wild type and NOD2^{-/-} mice. There long list of concerns compromises the ability of the authors to make conclusions possible connections to Crohn's disease.

The following constitutes a list of major/moderate concerns:

1. For studies related to Figure 1, the authors should stain for anti-SIgA (or anti-SC), not anti-IgA alone, to make claims about retrotranslocation. From the present figure, one cannot make claims about whether the IgA was enriched from inside (interstitial fluids) or outside (retrotranslocation). It is not implausible that interstitial IgA accumulates in the PP of CD patients.
2. For studies related to Figure 2, the authors should specify the number of PP used per experiment. The legend suggests 6 mice per experiment. Does that translate to one PP per mouse? That is an extremely small sample size.
3. For studies related to Figure 2, only SIgA was used for transport/uptake experiments. The authors should use an inert antigen (e.g., BSA) and a control immunoglobulin (e.g., IgG) to test whether the effects observed are specific (or not) to SIgA. The same applies to the oral immunization studies, which could have been done with IgG-p24 complexes or deglycosylated SIgA (as done previously by the a
4. The results in Figure 3 A should be replotted and analyzed statistically to compare wild type mice to the NOD2 KO mice for common treatment condition. For example, wild type and NOD2 mice for the DSS + laminarin should be plotted side by side and then statistics applied between the two mouse strains, not among treatments for single mouse strain as is currently presented. Only then will they authors be able to make claims about NOD. Also, there is no indication of sample sizes for these experiments.
5. On line 157-158, the authors state that Salmonella challenge in the presence of Sal4SIgA worsens DAI. This is counter intuitive since that antibody and similar work by the authors has shown that anti-LPS SIgA are protective and promote agglutination in the lumen. Are the authors making the claim that the NOD2 mice take up SIgA-aggregates? This should be shown by microscopy.
6. The transcytosis studies with Caco-2 with M-like cells conversion are lacking important controls, including the appearance /demonstration of M-like cells and that transcytosis occurs exclusively via these cells following conversion of Caco-2 monolayers. These controls should be included as supplemental figures.
7. In Figure 4, the knockdown studies should be accompanied by an apical-to-basolateral transport control with a protein known to use those relevant pathways (e.g., a toxin). Co-localization studies by confocal microscopy are also required. Simply demonstrating the effect of a knock down on SIgA transport by ELISA (and with a few select pulldown assays) is not sufficient to make claims about cellular pathways of transport.

Resubmission to *Nature Communications*

Manuscript number: NCOMMS-19-15971

Saint-Etienne, 20th December 2019

Dear Editor and Reviewers,

Thank you for taking the time to review this article. We greatly appreciate your work to improve our manuscript. We have taken into account your comments and areas for improvement. We sincerely hope that this revised version is clearer and will meet your expectations. Please find below a point-by-point response to each of the referee comments and a description of changes made.

Reviewer #1 (Remarks to the Author)

In the manuscript by Rochereau et al., the authors investigated whether NOD2 plays a role in IgA retrograde transport and whether it may influence the mucosal inflammation involved in Crohn's disease. As NOD2 is the strongest genetic risk factor for CD, this data is of interest to researchers studying host-microbe interactions and their involvement in the etiology of IBD. The authors identified a higher number of SIgA+ cells within ileal PP biopsies of CD patients with NOD2 mutations, as compared to CD without NOD2 mutations and healthy controls. This is the most intriguing result and initiated the remainder of the study investigating whether NOD2 is involved in IgA-complex reverse transcytosis. The IgA complexes co-localized with M cells and DCs, indicating the path via M cells and then taken up by dendritic cells within the PPs. The authors then demonstrated that a similar result could be observed in Nod2^{-/-} mice. The authors use an HIV protein model to further indicate a role for Nod2 in this IgA-complex immune response and while there was increased antigen-specific fecal IgA and serum IgG in Nod2^{-/-} mice, no cytokine differences (except for IL-10) were observed between WT and Nod2^{-/-} mice suggesting no difference in immune response.

The authors nicely show an effect of the response to Salmonella-IgA complexes as compared to DSS and Salmonella challenge, but it is not clear what the strength of the Nod2 involvement is since the Nod2KO mice were not statistically compared to the WT littermate mice receiving the same treatment. Were the WT and Nod2KO mice co-housed or separated at weaning? If the effect of the Salmonella-SIgA is similar in both WT and Nod2KO mice (as it appears in Fig 3), then it would be difficult to determine the exact influence of NOD2 on this phenotype.

This point is now detailed in the first major point of the reviewer 1.

The in vitro data showing increased SIgA transport in NOD2 knockdown cells and cells stimulated with MDP, provide strength to the initial findings of increased IgA+ cells in CD patients. Further, increased Dectin-1 and Siglec-5 expression in the knockdown cells and M-cells reinforce the involvement of NOD2 in IgA retrograde transport. Overall, the findings in CD patients with NOD2 mutations and data in Figure 5 present convincing evidence of a role for Nod2 in IgA-complex transport. This data would add to the field and should be considered for publication.

We greatly appreciate the positive appraisal of our work by Reviewer 1.

However, some questions remain:

NOD2 is an intracellular receptor – so would deletion in epithelial cells only (using a villin-cre mouse) lead to the same phenotype? Or does NOD2 need to be deleted in all cells involved in the IgA retrograde transport – e.g. also in plasma cells, dendritic cells?

IgA retrograde transport is only mediated via M cells (Rochereau *et al.* Plos Biology 2013). We deleted NOD2 in a specific *in vitro* M-like cells model containing only enterocytes and M cells. Using this specific model of FAE, we were able to monitor the role of NOD2 during IgA reverse transcytosis only in M cells.

Besides as the editor says “Exploration of cell type-specific roles of Nod2 is not a critical requirement from the editorial perspective.”, we didn’t went into cell-type specific roles.

*Since the mice used were littermates, it would be important to confirm that the gut microbiota is not significantly different in the mice used in Fig 2 to show increased IgA transport into the PP. Alternatively, comparing the reactivity of the IgA between WT and Nod2KO mice to the gut microbiota (similar to what was done in McCarthy *et al.* 2011 JCI) may further indicate when the role of NOD2 is most critical.*

We totally agree with this comment and have now performed these experiments as suggested by the reviewer. First, we verified the ability of IgA to bind to the same microbiota in littermate or NOD2 2KO mice, as previously described by McCarthy *et al* (Fig 2c). To quantify our observations, we also measured the MFI of IgA coated bacteria by flow cytometry (Fig 2b).

Major points:

*1. The initial findings and data in Figure 5 present convincing evidence of a role for NOD2 in IgA-complex retrograde transport, however, the *in vivo* data is underwhelming and does not provide direct evidence of a role for Nod2 in the sensing or immune response to these IgA-complexes. As this is the title and main conclusion of the paper, it is a concern. Could the data from Fig 2 (**we think that you talked about the figure 3 as you mentioned previously**) be represented to show WT and Nod2KO mice on the same plot to allow for statistical comparison? This is the only way to clearly show a role for Nod2 in the mice.*

We agree with the reviewer 1 and we replotted the salmonella-IgA without laminarin conditions for littermate WT and NOD2 KO mice only (supplemental Fig. 1d). Statistical analysis of this new comparison revealed that there was a bias in our first analysis. All inflammatory parameters measured in the serum (IL-6, CRP and LPS) were taken at D5 post-infection but for neutrophil infiltrations, the colon histology was done at the time of mouse sacrifice (D9 for WT mice as they showed fewer clinical signs and D5 for NOD2 KO mice). Weight loss from D5 to D9 of WT mice indicates that they became increasingly sick. The Nancy score was not comparable between the WT and NOD2 KO mice. We reproduced the experiment on littermate WT and NOD2 KO mice focusing on the laminarin-free condition. In addition to colon histological analysis, we also measured IL-6, LPS and CRP at D5 in the blood. With these new data presented in supplemental figure 1d and implemented in figure 3a, 3b, and supplemental figure 1a and 1b, we now confirm the role of NOD2 in our observations.

Minor point:

1. What is the reference for line 106-107? Is this only in mice, or also in humans deficient for NOD2?

The reference is indicated one sentence after, and it has been proven only in mice yet (Barreau F et al. PLoS One. 2007).

Reviewer #2 (Remarks to the Author):

In this manuscript, Rochereau and colleagues investigate the role of NOD2 in influencing the retrotranslocation of SIgA and SIgA-immune complexes into mouse (and a limited number of human) Peyer's patch tissues. While the study is intriguing, the investigators are quick to draw conclusions without sufficient experimental numbers and/or without important controls to justify claims about specificity of the SIgA transport in wild type and NOD2^{-/-} mice. Their long list of concerns compromises the ability of the authors to make conclusions possible connections to Crohn's disease.

The following constitutes a list of major/moderate concerns:

1. For studies related to Figure 1, the authors should stain for anti-SIgA (or anti-SC), not anti-IgA alone, to make claims about retrotranslocation. From the present figure, one cannot make claims about whether the IgA was enriched from inside (interstitial fluids) or outside (retrotranslocation). It is not implausible that interstitial IgA accumulates in the PP of CD patients.

We thank the reviewer 2 for this point. Colocalization between SIgA and DC-SIGN (Fig. 1b) was already added and clearly show that counted SIgA-positive cells could come from a retrograde transport of IgA through M cells and a consecutive uptake by dendritic cells. A new colocalization (Fig. 1b) using anti-IgA and anti-secretory component (SC) staining has been now added and confirmed that IgA was not enriched from inside (interstitial fluids) but from the lumen (retrotranslocation).

2. For studies related to Figure 2, the authors should specify the number of PP used per experiment. The legend suggests 6 mice per experiment. Does that translate to one PP per mouse? That is an extremely small sample size.

This experiment was repeated on 6 mice per group but each point represents the average of 3 PP per mouse. This precision has been added in the corresponding figure legend.

3. For studies related to Figure 2, only SIgA was used for transport/uptake experiments. The authors should use an inert antigen (e.g., BSA) and a control immunoglobulin (e.g., IgG) to test whether the effects observed are specific (or not) to SIgA.

We have now performed these control experiments (Fig 2a).

The same applies to the oral immunization studies, which could have been done with IgG-p24 complexes or deglycosylated SIgA (as done previously by the authors).

The role of IgA as a vector carrying a protein such as p24 has already been published (Rochereau et al, EJI 2014 and JACI 2015). We also showed that IgG was not able to cross epithelium via M cells *in vitro* (Rochereau et al, Plos Biology 2013) and *in vivo* (in this article). In Figure 2d, we already showed control groups such as IgA alone and p24 alone. Moreover, we know that administration of IgG-p24 by oral or nasal route does not induce p24-specific antibodies in WT mice (data not shown).

4. The results in Figure 3 A should be replotted and analyzed statistically to compare wild type mice to the NOD2 KO mice for common treatment condition. For example, wild type and NOD2 mice for the DSS + laminarin should be plotted side by side and then statistics applied between the two mouse strains, not among treatments for single mouse strain as is currently presented. Only then will they authors be able to make claims about NOD.

We replotted the salmonella-IgA without laminarin conditions for littermate WT and NOD2 KO mice only (supplemental Fig. 1d). Groups of pretreated mice with laminarin were not replotted as no colitis was observed.

Also, there is no indication of sample sizes for these experiments.

We agree and have now indicated these experimental details in the figure legend. “The experiment was repeated twice with 3 mice for each group showing similar results.”

5. On line 157-158, the authors state that Salmonella challenge in the presence of Sal4SIgA worsens DAI. This is counter intuitive since that antibody and similar work by the authors has shown that anti-LPS SIgA are protective and promote agglutination in the lumen. Are the authors making the claim that the NOD2 mice take up SIgA-aggregates? This should be shown by microscopy.

This is a very good remark. We have now performed agglutination experiments. Salmonella-IgA immune complexes were obtained under *in vitro* binding conditions and Salmonella aggregates formed by IgASal4 has now been quantified before oral challenge. Under our *in vitro* binding conditions, IgASal4-bound Salmonella Typhimurium (SL1344 strain) do not form aggregates (Supplementary Fig. 1c). So contrary to what was done in the study from Mantis et al. (*Immunol Invest.* 2010), IgASal4, in our *in vitro* binding condition, with our Salmonella Typhimurium strain (SL1344), do not form aggregates. We were not able to measure aggregate *in vivo*, so we can't exclude that NOD2 KO mice were able to uptake potential SIgA-aggregates formed *in vivo* after oral administration due to the increase activity of reverse transcytosis.

6. The transcytosis studies with Caco-2 with M-like cells conversion are lacking important controls, including the appearance /demonstration of M-like cells and that transcytosis occurs exclusively via these cells following conversion of Caco-2 monolayers. These controls should be included as supplemental figures.

The M-like cells model that we used has been published and highly detailed previously (Rochereau *et al.* Plos Biology 2013). The presence of M cells has been shown by electronic microscopy, immunofluorescence staining, TEER measurement, passive transport of nanoparticles. Presence of tight junctions was observed between all cells. Moreover in order to see that IgA transcytosis occurs exclusively via M-like cells, all our experiments (including in this article) were conduct with the appropriate control which is a monolayer of caco-2 without Raji cells (so without M-like cells) and TEER is always checked before any transport experiments (see supplemental figure 2c). Here is an immunofluorescence staining we made to demonstrate the presence of M-like cells.

7. In Figure 4, the knockdown studies should be accompanied by an apical-to-basolateral transport control with a protein known to use those relevant pathways (e.g., a toxin).

We have now performed this control experiment using cholera toxin which is known to use Rab6 pathway (Supplementary Fig. 2e).

Co-localization studies by confocal microscopy are also required. Simply demonstrating the effect of a knock down on SIgA transport by ELISA (and with a few select pulldown assays) is not sufficient to make claims about cellular pathways of transport.

Thanks to have highlighted this point. Another confirmation of the interaction between IgA and these Rab proteins has been also observed by immunofluorescence (Fig. 4c).

We deeply hope that the point-by-point reply to the Reviewers, will provide convincing evidence of the importance of the role of SIgA reverse transcytosis in CD.

Sincerely,

Dr. Nicolas Rochereau, PhD

REVIEWERS' COMMENTS:

Reviewer #1 (Remarks to the Author):

The authors have responded satisfactorily to my comments.

Reviewer #2 (Remarks to the Author):

The authors (Rochereau et al) have done an excellent job at addressing my critiques concerning the need for appropriate controls for transcytosis assay, revealing sample sizes when appropriate, specifying IgA versus SIgA in Peyer's patch tissues and clarifying questions about uptake of Salmonella.